# Innate Immunity Crosstalk with *Helicobacter pylori*: Pattern Recognition Receptors and Cellular Responses

**DOI:** 10.3390/ijms23147561

**Published:** 2022-07-08

**Authors:** Yi Ying Cheok, Grace Min Yi Tan, Chalystha Yie Qin Lee, Suhailah Abdullah, Chung Yeng Looi, Won Fen Wong

**Affiliations:** 1Department of Medical Microbiology, Faculty of Medicine, University of Malaya, Kuala Lumpur 50603, Malaysia; heathercheok@gmail.com (Y.Y.C.); gtanmy89@gmail.com (G.M.Y.T.); chalystha@gmail.com (C.Y.Q.L.); 2Department of Medicine, Faculty of Medicine, University of Malaya, Kuala Lumpur 50603, Malaysia; suhailah73@um.edu.my; 3School of Biosciences, Faculty of Health & Medical Sciences, Taylor’s University, Subang Jaya 47500, Malaysia; chungyeng.looi@taylors.edu.my

**Keywords:** *Helicobacter pylori*, innate immune activation, pattern recognition receptors, TLRs, CLRs, NLRs, RLRs, macrophages

## Abstract

*Helicobacter pylori* is one of the most successful gastric pathogens that has co-existed with human for centuries. *H. pylori* is recognized by the host immune system through human pattern recognition receptors (PRRs), such as toll-like receptors (TLRs), C-type lectin like receptors (CLRs), NOD-like receptors (NLRs), and RIG-I-like receptors (RLRs), which activate downstream signaling pathways. Following bacterial recognition, the first responders of the innate immune system, including neutrophils, macrophages, and dendritic cells, eradicate the bacteria through phagocytic and inflammatory reaction. This review provides current understanding of the interaction between the innate arm of host immunity and *H. pylori*, by summarizing *H. pylori* recognition by PRRs, and the subsequent signaling pathway activation in host innate immune cells.

## 1. Introduction

*Helicobacter pylori* is a microaerophilic, Gram-negative bacterium that colonizes the gastric epithelium of roughly half of the world’s population [1,2]. Although its infection remains asymptomatic in the majority of infected individuals, some patients develop gastroduodenal pathologies that progress into gastritis, peptic ulcer, gastric adenocarcinoma, and mucosa-associated lymphoid tissue lymphoma [3,4]. The unique characteristics of *H. pylori* contribute to its efficient infection in human hosts. For instance, the helical shape of the bacteria, its ability to produce urease, and its flagella-dependent corkscrew movement pattern allow bacterial invasion and colonization at the unconducive gastric mucosal layer. The release of virulence proteins, such as vacuolating cytotoxin A (VacA) or cytotoxin-associated gene A (CagA) through type IV secretion system (T4SS), causes damage to the gastric epithelial cells and initiates the inflammatory reaction that precedes gastritis and gastric carcinoma.

The host innate immune system is the first layer of defense against foreign pathogen invasion. *H. pylori* infection triggers activation of neutrophils, macrophages, and dendritic cells through the recognition of the bacteria motifs via innate immune receptors [5]. Interestingly, evidence has increasingly revealed the exploitation of immune cells and pattern recognition receptors (PRRs) to circumvent immune activation during *H. pylori* infection [6,7]. This warrants further understanding on the interaction between host immune response and *H. pylori*.

This current review discusses the interaction of *H. pylori* with the host innate immune system, from the molecular recognition by PRRs to the response of innate immune cells toward *H. pylori*. This helps in the comprehension of *H. pylori*–host interactions based on the latest literature, hence providing possibilities for developing effective treatments and prophylactic strategies in the future.

## 2. Immune Recognition of *H. pylori*

Immune activation elicited by *H. pylori* begins with the recognition of the bacterial-pathogen-associated molecular patterns (PAMPs) by PRRs that are expressed by the innate immune cells or gastric epithelial cells. The major classifications of PRRs, including toll-like receptors (TLRs), NOD-like receptors (NLRs), C-type lectin receptors (CLRs), and retinoic acid-inducible gene (RIG)-I-like receptors (RLRs), are all involved in *H. pylori* recognition and innate immune activation [8,9,10]. Figure 1 depicts the interaction of different PRRs with *H. pylori* ligands and their downstream signaling pathway activation.

### 2.1. Toll-Like Receptors (TLRs)

TLRs are a family of type I transmembrane proteins. Each receptor harbors a leucine-rich repeat ectodomain, a transmembrane region involved in PAMP recognition, and an intracellular Toll-IL-1 receptor (TIR) domain to mediate signal transduction. Upon binding to PAMPs, TLRs form complexes and activate either myeloid differentiation primary response 88 (MyD88)-dependent or -independent cascade to switch on nuclear factor kappa B (NF-κB) or interferon regulatory factors (IRFs) transcription factors, thus initiating an immune reaction to counter the pathogen invasion [11]. There are a total of 10 TLRs in humans, some of which (TLR1, TLR2, TLR4, TLR5, TLR6, and TLR10) are present on cellular plasma membranes to recognize extracellular microbial components such as lipopolysaccharide (LPS), lipoproteins, and peptidoglycans; others (TLR3, TLR7, TLR8, and TLR9) are attached within the lipid bilayer of the intracellular vesicles to recognize microbial nucleic acids. 

TLR4 is the most intensively studied member of the TLR family. It recognizes *H. pylori* LPS [11] and activates the NF-ĸB transcription factor, which triggers production of pro-inflammatory cytokines, including interleukin-1β (IL-1β), IL-2, IL-6, IL-8, and IL-12 [12]. However, some studies denounce the role of TLR4 in immunity to *H. pylori* [13,14,15,16]. Instead, TLR2 was suggested as the receptor for *H. pylori* LPS despite its common role for detecting bacterial lipoproteins [17]. Together with TLR10, the TLR2/TLR10 heterodimer elicits NF-κB activation following exposure to heat-inactivated *H. pylori* or LPS [7,18]. *H. pylori* LPS-bound TLR2 also interacts with TLR4 to promote gastric metaplasia, dysplasia, and adenocarcinoma [19,20]. 

TLR5 is well-known for its recognition of bacterial flagellin, and is constitutively expressed on the surface of epithelial cells as well as some innate immune cells [21]. Previous research showed that live *H. pylori* or its purified flagellin activates the NF-κB pathway through binding to TLR5 [13]. Significant upregulation of TLR5 is also detected in THP-1 cells following *H. pylori* infection, causing cytokine secretion of IL-8 and TNF-α, which initiate inflammation [22]. Recent studies have reported that components of T4SS, such as CagL and CagY, in pathogenic *H. pylori* strains can serve as TLR5 agonists in driving the innate immune activation and T helper 1 (T_H_1) cells’ recruitment [23,24].

Inside the intracellular vesicles, TLR7 and TLR8 recognize *H. pylori* RNA [13], whereas TLR9 recognizes unmethylated CpG DNAs. TLR8 was shown to be induced in THP-1 monocytic cells following the phagocytosis of *H. pylori* [25], whereby an absence of MyD88 abrogated the production of cytokines such as IL-6 and IL-12 following *H. pylori* RNA stimulation [26]. *H. pylori* or its purified DNA induces secretion of IL-8 in neutrophils. Depletion of TLR9, by neutralizing antibodies or inhibitory oligonucleotide, abrogates IL-8 production, suggesting its vital involvement in the bacterial recognition and cell activation [27]. Intriguingly, another study indicated an anti-inflammatory role of TLR9, possibly through the secretion of type I interferon (IFN) in the early stage of *H. pylori* infection-mediated gastritis [28]. Mice deficient in TLR9 display an intensively augmented inflammation and IL-17 production when infected with *H. pylori* strains with intact T4SS, an important channel for transferring bacteria DNA into host cells [29]. It was reported that TLR9-mediated transcriptional activity can differ depending on cellular polarity, in which basolateral TLR9 signals IkappaB (IκB) degradation and activation of the NF-κB pathway, whereas apical TLR9 inhibits NF-κB activation. Hence, we anticipate that the anti-inflammatory response invoked by TLR9 signaling following *H. pylori* infection may be attributable to bacterial-T4SS-mediated immune suppression, which diverts TLR9 to the apical surface to promote a tolerogenic response. 

### 2.2. C-Type Lectin Receptors (CLRs)

CLRs belong to an important class of PRRs that bind to carbohydrate moieties using a conserved carbohydrate-recognition domain (CRD). In general, CLRs induce downstream signaling via two pathways: one involves immunoreceptor tyrosine-based activation motif (ITAM)-containing adaptor molecules such as Fc receptor γ-chain (FcRγ); the other involves phosphatases or kinases that either directly or indirectly interact with cytoplasmic domain of the receptor [30,31]. 

Among CLR family members, dendritic-cell-specific intercellular nonintegrin (DC-SIGN) plays a pivotal role in *H. pylori* recognition and pathogenesis. DC-SIGN interacts with mannose and fucose moieties, and is capable of releasing signals to modulate TLR signaling. A study by Gringhuis, et al. [32] demonstrated that *H. pylori*, a fucose-expressing bacteria, inhibits T_H_1 polarization by enhancing IL-10 and decreasing IL-12 production. Manipulation of DC-SIGN by *H. pylori* also extends to their ability to spontaneously switch the expression of Lewis antigens on LPS on or off, whereby the presence of Lewis antigenic expression triggers DC-SIGN on dendritic cells to suppress T_H_1 polarization and vice versa [33]. 

In addition to DC-SIGN, the expression of macrophage-inducible C-type lectin (MINCLE) is also upregulated in THP-1 human monocytic cell line during direct or indirect infection with *H. pylori*. Lipid cholesteryl acyl α-glucoside (αCAG) and cholesteryl phosphatidyl α-glucoside (αCPG) are components of *H. pylori* that bind to MINCLE to induce secretion of proinflammatory cytokines such as tumor necrosis factor (TNF) and macrophage inflammatory protein (MIP) in dendritic cells, hence exacerbating gastritis in the host [34]. The interactions between MINCLE with the Lewis antigens of *H. pylori* LPS, on the contrary, induces an anti-inflammatory response in macrophages [35]. Taken together, *H. pylori* exploits both DC-SIGN and MINCLE to evade the host immune response while inflicting chronic inflammation.

### 2.3. NOD-Like Receptors (NLRs) 

The human nucleotide-binding and oligomerization domain (NOD)-like receptor (NLR) family consists of 22 members that share common features: an N-terminal effector domain to mediate signal transduction, a centrally located nucleotide-binding domain (NBD/NATCH) for oligomerization, and a C-terminal leucine-rich repeat (LRR) domain for ligand sensing. NLRs are involved in the formation of inflammasome, which recruits adaptor protein apoptosis-associated speck-like protein containing a CARD (ASC) to cleave effector molecule pro-caspase-1 into active caspase-1 and mediate secretion of pro-inflammatory cytokines IL-1 and IL-18. Some NLRs, such as NOD1 and NOD2, signal mitogen-activated protein kinase (MAPK) and NF-κB in an inflammasome-independent manner [36,37]. 

Gene expression analysis showed elevated NLRs levels, including NLRC4, NLRC5, NLRP9, NLRP12, and NLRX1, in the *H. pylori* challenged human monocytic THP1 cells [38]. Kim, et al. [39] discovered that NOD2, but not NOD1, is required for the activation of NLRP3 inflammasome and pro-IL-1β production in *H. pylori* challenged bone-marrow-derived dendritic cells.

The production of mature IL-1β in dendritic cells can be triggered by *H. pylori* virulence factor cytotoxin-associated gene pathogenicity island and CagL [39]. However, NLRP3 activation following acute *H. pylori* infection results in low levels of mature IL-1β secretion, despite producing abundant premature pro-IL-1β, suggesting *H. pylori* interference with NLRP3 inflammasome activity to avoid host immune reactions [40]. A deficiency of caspase-1 and IL-1β, but not NLRP3, causes impairment of bacteria clearance from the stomach in *H. pylori* infected mice [39], which may be due to the involvement of another pathway in the cleavage of caspase-1, which is necessary for IL-1 maturation [41].

Interestingly, genetic polymorphisms involving different NLR members or pathway molecules (NLRP3, NLRP12, NLRX1, and CARD8) are associated with the incidence of gastric cancer. For instance, NLRP12 rs2866112 polymorphism increases the risk of *H. pylori* infection [38]. More studies are required to determine the involvement and function of these different members of NLRs family in *H. pylori* recognition. Together, these studies indicate the participation of NLRs in *H. pylori* recognition and immune cell activation through the release of IL-1.

### 2.4. RIG-I-Like Receptors (RLRs)

RLRs consist of three members known as retinoic acid-inducible gene I (RIG-I), melanoma differentiation association gene 5 (MDA5), and laboratory of genetics and physiology 2 (LGP2), which detect intracellular RNAs to trigger MyD88-independent secretion of type 1 interferon (IFN) [42]. 

It was reported that *H. pylori* 5′-triphosphorylated RNA can be recognized by the RIG-I receptor [26]. HEK293 cells transfected with RIG-I expressing plasmid activate the IFNβ promoter in response to *H. pylori* RNA stimulation. RIG-1 recognition of *H. pylori* RNA occurs in a phosphate-dependent manner, as a prior RNA dephosphorylation or usage of RIG-I K858E mutant, which cannot interact with 5’-triphosphate RNA, abrogates cell activation [26]. 

Furthermore, a clinical study suggested that expression of RIG-1 significantly correlates with poor gastric cancer prognosis [43]. Knockdown of RIG-1 using RNA interference technology in human gastric cancer cell lines results in augmented cell migration, enhanced division in G2/M phase, and higher cell viability [43], suggesting RIG-1 as a crucial component for the prevention of cancer cell growth and invasion. A study using human gastric biopsies revealed a significant upregulation of MDA5 expression in *H. pylori* infected samples, whereby the level of expression correlated with clinical parameters of gastric atrophy and intestinal metaplasia [44]. 

## 3. Innate Immune Cell Activation and Suppression by *H. pylori*

The immune response against *H. pylori* is initiated across gastric epithelial cells (GECs), either through the signaling or recruitment by activated GEC or through the direct recognition of various bacterial components or virulence factors [45]. This causes a cascade of innate immune cell activation by neutrophils, macrophages, and dendritic cells. Nonetheless, the host innate immune response can be a double-edged sword in the context of *H. pylori* infection, where it controls both the bacterial load in the gastric mucosa, as well as the proinflammatory responses that lead to gastritis. Hence, polymorphonuclear cells infiltration has been long used in histological analysis as an indicator of gastritis severity [46]. Figure 2 summarizes the mechanism utilized by *H. pylori* to promote inflammation and enhance survival, as discussed in the following subsections.

### 3.1. Neutrophils

The extensive neutrophil recruitment and activation during *H. pylori* infection lead to an excessive degree of inflammation and mucosal damage [47]. The peripheral blood neutrophil to lymphocyte ratio was shown to be lower in *H. pylori* positive patients compared with the uninfected individuals [48]. Neutrophils are highly sensitive to the presence of *H. pylori*, mainly due to the secretion of the *H. pylori* virulence factor neutrophil activating protein (HP-NAP), which serves as a chemotactic factor [49,50]. Neutrophils rapidly secrete cytokines IL-8, IL-1β, and TNF-α following *H. pylori* infection [51]. Additionally, *H. pylori* infection induces hepatoma-derived growth factor (HDGF) expression to enhance neutrophilic infiltration, which is pivotal in gastric carcinogenesis [52]. HDGF ablation in a mouse model blocked neutrophil recruitment following *H. pylori* infection, and demonstrated an alleviation of gastric lesions [52]. The presence of a flagellin A (FlaA)-positive strain, and a subunit of the T4SS, CagL, induce the secretion of the proinflammatory cytokine IL-1β from neutrophils [53]. 

The activation of neutrophils can be dampened by *H. pylori* to promote bacterial persistence. Cathepsin C expression was implicated in playing such a role, whereby cagA-positive *H. pylori* strain is capable of inhibiting cathepsin C in gastric epithelial cells to prevent neutrophils infiltration and bacterial clearance [54]. In addition, *H. pylori* is able to activate specific cellular carcinoembryonic antigen-related cell adhesion molecule (CEACAM) in order to promote bacterial uptake by neutrophils, while supporting their intracellular survival through CEACAM–*H. pylori* outer membrane adhesin HopQ [55]. 

### 3.2. Monocytes/Macrophages

Monocytes and macrophages serve as phagocytes and antigen-presenting cells upon *H. pylori* infection. They are responsible for the polarization of T_H_1 via secretions of IL-12 and IL-23 [56]. The importance of macrophages in *H. pylori* infection is highlighted by their marked increases in inflammatory markers, cell motility [57], and infiltration into the peritoneum as early as 2 days post-infection [58]. Furthermore, transient removal of CD11b^+^ macrophage population was identified to reduce gastric pathology in *H. pylori* infected mice without altering the bacterial load, implying its critical role in gastric pathology [59]. The activation of macrophages also relies on bacterial factors, whereby the lipopolysaccharide (LPS) inner -core heptose metabolites (ADP-heptose) reportedly induce NF-κB activation and early maturation of the cells [60]. 

Recently studies highlighted the involvement of the Notch signaling pathway in *H. pylori* challenged macrophages [61]. The upregulation of Notch ligand Jagged1 was detected in macrophages co-cultured with *H. pylori*, and clinical specimens from *H. pylori* infected patients displayed higher Jagged1+ macrophages [61]. The overexpression of Jagged1 in macrophages promotes bactericidal activity through facilitating pro-inflammatory mediator release. Conversely, siRNA-mediated knockdown or anti-Jagged1 neutralizing antibodies attenuate these activities. Additionally, γ-secretase, an inhibitor of Notch signaling was shown to suppress the function of macrophages against *H. pylori* infection through nitric oxide (NO) production, proinflammatory cytokine secretion, phagocytosis, as well as bactericidal activities [61]. 

Polarization of macrophages remains an important topic during *H. pylori* infection. The classically activated macrophages (M1) display a vigorous proinflammatory response and phagocytic function, leading to the reduction in bacterial load and enhanced pathology; whereas alternatively activated macrophages (M2) exhibit anti-inflammatory and immune-modulating functions during tissue remodeling and healing. It is evident that M1 polarization is prominent in *H. pylori* infected C57BL/6 mice, and can be further enhanced with vaccination using the sublingual administration of *H. pylori* lysate, although a mixed M1/M2 phenotype is commonly reported in human infection settings [62,63]. 

Factors influencing the polarization and subsequent outcome of infection have been identified in various reports, including the expression of the cation channel transient receptor potential melastatin 2 (TRPM2). The absence of TRPM2 promotes M1 polarization in response to *H. pylori* infection. Mice deficient in TRPM2 demonstrate higher gastric inflammation and lower bacterial colonization compared with control mice, owing to overloading calcium levels in macrophages. High intracellular calcium levels elicit an enhanced MAPK signaling pathway and increased production of iNOS and IL-6, which perpetuate inflammation for the resolution of bacterial infection [64]. 

Another study highlighted the importance of matrix metalloproteinase 7 (MMP7) during polarization. MMP7 null mice displayed increased M1 cells that synthesized more IL-1β than the wildtype counterpart during both in vivo and ex vivo infection with *H. pylori* [65]. Increased incidences of gastric hyperplasia and dysplasia were detected when MMP was depleted in hypergastrinemic mice due to increased M1 macrophage polarization and mucosal inflammation [65].

Apart from host factors, bacterial virulence, such as CagA, was also shown to influence the M1/M2 balance. The phosphorylation of CagA increases gene expression encoding heme oxygenase-1, causing macrophage polarization toward the M2 phenotype, which functions poorly in bacterial elimination [66]. Taken together, the interplay between various host and bacterial factors is crucial in controlling the outcome of infection, targeting at achieving optimal activation for bacterial clearance (M1) and sufficient healing to prevent pathology (M2).

To achieve long-term survival in the host, *H. pylori* develops strategies to impede macrophage functions. One example is the bacterial ability to inhibit macrophage phagocytosis [67,68] and to induce macrophage arginase II (ARG2) to inhibit nitric oxide (NO) production [69,70]. Additionally, *H. pylori* disrupts proliferation [71] and causes the apoptotic program by promoting hydrogen peroxide release and mitochondrial membrane depolarization [72,73]. Thus, the crosstalk between bacteria and macrophages is important to determine if infection can be circumvented in the host.

### 3.3. Dendritic Cells

*H. pylori* infection induces dendritic cells’ activation and secretion of cytokines IL-10, IL-12, IL-6, and IL-8 in a dose-dependent manner [74]. It was identified that human HLA-DR^+^ gastric dendritic cells and monocytes-derived dendritic cells engulf the bacteria for antigen presentation to CD4^+^ T cells, which preferentially skews T cells into T_H_1 cells, which release interferon-γ (IFN-γ) [75]. The induction of T helper 17 cells (T_H_17) is also present alongside T_H_1 cells, as shown in other reports [76,77]. *H. pylori cagPAI* influences the maintenance and development of T_H_1 and T_H_17 cells through augmenting the secretion of IL-12 and IL-23 from dendritic cells [76,77].

In *H. pylori* infected pediatric patients, an increased expression of the high-affinity IgE receptor (FcεRI) was observed on peripheral myeloid and plasmacytoid dendritic cells [78]. Monocyte-derived dendritic cells exhibit high FcεRI and IL-10 expressions, suggesting *H. pylori* drives regulatory dendritic cell differentiation to diminish the hostile immune environment [78]. The immunomodulatory role of dendritic cells was supported by Russler-Germain, et al. [79], whose demonstrated that migratory classical dendritic cells promote peripheral regulatory T cell differentiation to exert immune tolerance. Programmed cell death ligand 1 (PD-L1)-expressing dendritic cells interact with PD-1-expressing T cells in gastritis lesions to impede *Helicobacter* induced inflammation, thus warranting persistent *Helicobacter* colonization in mice [80]. Additionally, *H. pylori* infection can impair the antigen-presentation function of dendritic cells to T cells, thereby blocking T_H_1 differentiation [81].

## 4. Conclusions

*H. pylori*–mediated immune activation begins with the recognition of bacterial PAMPs by gastric epithelial or immune cell PRRs. Upon recognition, a cascade of immune activation is made possible for bacterial clearance as well as chronic gut inflammation. Although multiple studies have helped to deepen our understanding of host–*H. pylori* interactions, more remains to be investigated to explain the different outcomes of *H. pylori* infection. Recent studies have focused on identification of various PRRs-associated risk alleles [82], which would provide a better understanding on the risk of infection and degree of severity. 

## Figures and Tables

**Figure 1 ijms-23-07561-f001:**
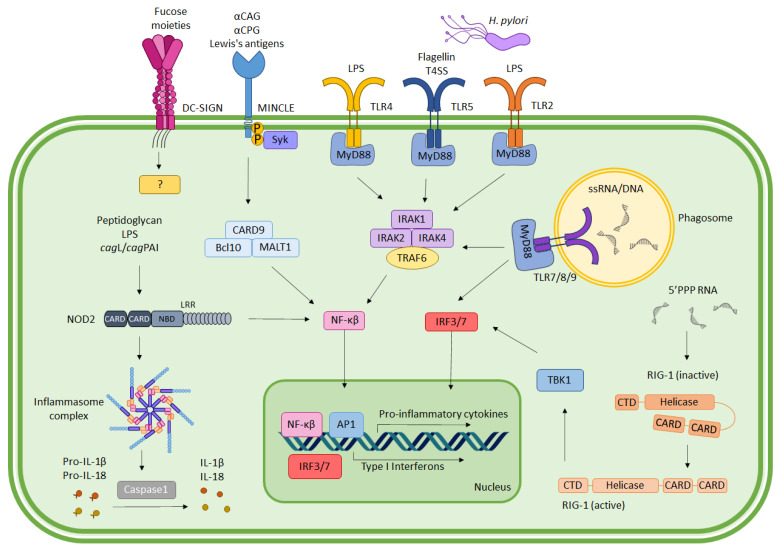
Activation of pattern recognition receptors (PRRs) by *H. pylori*. Toll-like receptors (TLRs) 2, 4, and 5 recognize different extracellular bacterial factors to mediate MyD88-dependent or -independent activation of proinflammatory cytokine secretion. TLR7/8/9 recognize ingested bacterial DNA or single-stranded RNA (ssRNA) in phagosomes for activation of type I interferons production. Transcription factors NF-ĸB, activator protein 1 (AP1), and IRF3/7 play prominent roles in transactivation of inflammatory cytokines and type I interferons. C-type lectin-like receptors (CLRs) including MINCLE also induce NF-ĸB through interaction with CARD9/BCL10/MALT1. DC-SIGN transmits signals through an unknown pathway during *H. pylori* infection. NOD-like receptors (NLRs) such as NOD2 activate the inflammasome complex to induce production and maturation of IL-1β and IL-18. RIG-I-like receptors (RLRs) such as retinoic acid-inducible gene I (RIG-I) are activated by 5′triphosphate (PPP) single-stranded RNA, leading to conformation change, which subsequently activates Tank-binding kinase 1 (TBK1). This, in turn, causes production of type I interferons through IRF3/7 signaling.

**Figure 2 ijms-23-07561-f002:**
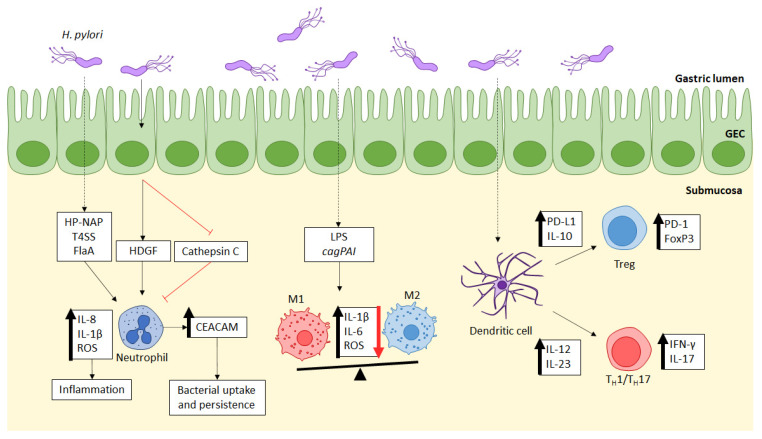
Innate immune cells interaction with *H. pylori* across gastric epithelial cells (GECs). Neutrophils are directly activated by bacterial antigens, including *H. pylori* neutrophil activating protein (HP-NAP), type IV secretion system (T4SS), and flagellin A (FlaA), or indirectly through gastric epithelial cells secretion of hepatoma-derived growth factor (HDGF). Activation may result in proinflammatory cytokines secretion (IL-8 and IL-1β) and the production of reactive oxygen species (ROS) for the clearance of bacterial load. *H. pylori* can inhibit cathepsin C production to dampen neutrophil activation. It can also lead to cellular carcinoembryonic antigen-related cell adhesion molecules (CEACAMs) expression, which enhances neutrophils phagocytic activity without bacterial killing. Macrophages can be activated by bacterial lipopolysaccharide (LPS) or by the presence of the virulence gene cag pathogenicity island (cagPAI), resulting in mixed polarization of proinflammatory (M1) with high expressions of IL-1β, IL-6, and ROS, and the anti-inflammatory (M2) macrophages. Dendritic cells process bacterial antigens that are presented to T cells for differentiation. These cells upregulate programmed death ligand 1 (PD-L1) on regulatory T cell (Treg) to increase IL-10 secretion, leading to bacterial persistence. It can also upregulate proinflammatory cytokines IL-12 and IL-23 to stimulate T helper 1 (T_H_1) and 17 (T_H_17) polarization, which releases high amounts of interferon-γ (IFN-γ) and IL-17.

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
