# Peer review of "Innate Immunity Crosstalk with Helicobacter pylori: Pattern Recognition Receptors and Cellular Responses"

_ijms, 2022, doi:10.3390/ijms23147561_

Round 1
Reviewer 1 Report
In this review titled " Innate immunity crosstalk with Helicobacter pylori: pattern recognition receptors and cellular responses", the authors compile a vast amount of published literature on the innate immune response to H. pylori infection with specific focus on PRRs and cellular responses.
The review is satisfactorily written and the illustrations are well thought out.
The language needs to be proof read properly. Minor mistakes like in line 46 "recognition of the bacterium motifs..." should be "recognition of bacterial motifs..." etc should be corrected.
A minor suggestion is that while the review is useful bringing existing knowledge together, the authors fail to add their input/thoughts on the gaps/contradictions in knowledge that remain. For example, TLR9 seems to be important for immune cell activation through IL8 induction and involved in anti-inflammatory signaling through IFN in H. pylori infection. Perhaps it would be helpful to include these thoughts more elaborately in the respective sections.
Author Response
We thank the reviewer for valuable comments to improve the manuscript. We have checked the language throughout the text, and minor mistakes (including that in line 46) have been corrected as advised. We have also added our discussion that plausibly explain the discrepancy of TLR9 data, as advised by the reviewer (please see page 3 line 126-135).
Reviewer 2 Report
The article represents a classic review work. The structure is clear and well thought out. The topic of the paper fits well with the aim of the journal. No substantive errors were noticed. The figures are imaginative and complement the information well. There is some confusion about the extent to which the work is innovative. It is a reliable compedium of knowledge in the field it describes. In the reviewer's opinion, one could be tempted to include some new information relating to the discussed bacterial-host interactions, which would go beyond the canon known so far. However, I leave the final decision in this respect to the Editor.
Author Response
We thank the reviewer for the generous comments given. We have added some new information to discuss the bacterial-host interaction (please see page 3 line 126-135), as advised.